# Research on the Construction of China's Provincial Food Security Evaluation System and Regional Performance—Based on the "Great Food View"

**Qijun Jiang [1], Zhijie Rong [1,*] and Zhiwei Yuan [2]**

[1] School of Economics and Management, Shanghai Ocean University, Shanghai 201306, China; qjjiang@shou.edu.cn
[2] School of Economics and Management, Dalian University of Technology, Dalian 116000, China; yuanzhiweisachiel@163.com
*   Correspondence: rongzj0806@163.com

**Abstract:** Based on the internationally recognized concept of food security and the scientific connotation of the "Great Food View". This research constructs China's provincial food security evaluation system under the "Great Food View" by comprehensively considering the regional food supply logic and dietary habits differences. Combining the improved entropy weight method and AHP to quantitatively evaluate the current situation and trend of food security in China's provinces (except Hong Kong, Macao, and Taiwan) from 2009 to 2021 by using this system—on the basis of dividing China's provinces into main food production provinces and non-main food production provinces. The results demonstrate that all provinces have entered the ranks of "security" and above food security, and the mismatch between "high vulnerability and low security" has been alleviated. Yet, the problem of "difficult growth" in the provinces with the lowest score of food security has emerged. The green and sustainability of the food security system in the main food production areas have been at a low level, and the food security in non-main food production areas has seen the "Matthew effect" of uneven development. Finally, policy implications are proposed from the aspects of diversified development of food sources, precision agriculture subsidy guide, optimization of farmland compensation and protection policies, and optimization of food storage and distribution.

**Keywords:** "Great Food View"; food security; index evaluation; entropy weight method; AHP method



## 1. Introduction

Food security risk is a long-term serious challenge in the process of human society development. Since 2020, global food insecurity has become more and more serious due to the COVID-19 epidemic, economic fluctuations, climate change, commodity price fluctuations, animal and plant diseases and pests, and regional conflicts. China is a country with relatively insufficient arable land per capita and water resources. For a long time, China has fed 20% of the world's population with 9% of the world's arable land and 6% of its freshwater resources. Therefore, China can ensure domestic food security, which is an important contribution to world food security. Yet now, the transformation of Chinese residents' food consumption structure [1] and the geographical mismatch between food production centers and consumption centers [2,3] have brought many new challenges to the food supply system [4]. For this reason, the *20th National People's Congress of China*—held every five years, China's most important conference—clearly proposed that "establishing a Great Food View" and "building a diversified food supply system" pointed out a new path for food security through the "Great Food View". However, no scholars have explored how China's domestic food security should be evaluated from the perspective gansuof the "Great Food View", and how the level of domestic food security is from the perspective of the "Great Food View". In conclusion, building a food security evaluation system

and scientifically evaluating the food security performance of different provinces under the "Great Food View" has important theoretical and practical significance for ensuring food security in China and the world, formulating reasonable food security policies, and continuing to improve the level of food security.

This paper closely follows the internationally recognized definition of food security and takes the new connotation of food security given by the "Great Food View" as the guidance to construct the evaluation system of China's provincial food security. It includes the differences in methods that safeguard food security between main and non-main food production areas, as well as the dietary habits of different regions, into the evaluation system to evaluate and measure China's provincial food security level. The innovation of this research mainly lies in:

First, the objective of previous research on evaluating China's food security level is "grain supply". This paper constructs an evaluation system from the "Great Food View" and focuses on "food supply" with nutrition and food acquisition as the guidance.

Second, in the past, the measurement of China's food security level was often only aimed at the main food production areas (including several main food production provinces) or the national macro level, ignoring the provinces in non-main food production areas (including several non-main food production provinces). This paper combines subjective and objective weighting methods to eliminate the impact of different safeguarding modes of food security in main food production areas and non-main food production areas and evaluates and measures the food security level of all provinces.

Third, different regions have different diet styles, and the micro elements of food security are also different. In the past, few studies on food security evaluation have included dietetic structure in the evaluation system. However, in this study, the dietary structure was included in the evaluation system from two aspects. First, according to the different dietary structures in different provinces, the corresponding adjustment was made in the calculation of indicators through objective methods to make them more realistic. The second is to scientifically calculate the rationality of dietary habits in different regions and incorporate the deviation between regional dietary structure and a standard healthy diet into the evaluation system.

## 2. Literature Review

### 2.1. Food Security and "Great Food View"

2.1.1. Authoritative Definition of Food Security

The definition of "food security" has been evolving with economic and social development since it was first proposed by the *Food and Agriculture Organization* (FAO) in 1974. *The World Food Security Summit* in 2009 proposed the most internationally recognized concept, and it has been used up to now [5], which is, "All people can obtain sufficient, safe and nutritious food at any time through material, social and economic means, meet their dietary needs and food preferences, and lead an active and healthy life".

2.1.2. Introduction of "Great Food View"

The concept of the "Great Food View" was discussed and determined at several national conferences in China. The "Great Food View" was first proposed at the *Central Rural Work Conference* in 2015. In 2016, the *No. 1 Document of the Central Committee of the Communist Party of China* took "building a Great Food View" as an important content to promote "Agricultural Supply Side Structural Reform"—one of the most important tasks in China's agricultural sector. In 2017, the *Central Rural Work Conference* further proposed to "establish a Great Food View" and "develop food resources in an all-round way". Now, in 2022, the *20th National People's Congress of China* was further emphasizing "building a Great Food View" and "building a diversified food supply system".

Academically, food security under the "Great Food View" should first be based on the basic national conditions of China with more people and less farmland, within the bearing range of resources and environment, take nutrition as the guidance, and coordinate the

"mountains, rivers, forests, fields, lakes, grass and sand" to develop multiple food sources. Second, food security under the "Great Food View" should grasp the changing trend of the people's food structure and promote the layout of green agricultural production that matches the market demand. Third, food security under the "Great Food View" should constantly strengthen the resilience of the food supply chain, give full play to the decisive role of the market in resource allocation, and break through the supply blockages connecting the large markets in non-main food production areas and the large resources in main food production areas [6,7].

### 2.1.3. The Relationship between Food Security and "Great Food View"

The relationship between the internationally recognized concept of food security and the concept of food security under the "Great Food View" is dialectical and unified. Further, the concept of food security under the "Great Food View" is the refinement of the internationally recognized concept of food security in China. The food security concept under the "Great Food View" not only emphasizes the balanced development and diversified acquisition of a variety of food products but also makes "nutrition, green and sustainable security" important indicators of food security in China and gives new elaboration to the international food security concept from the aspects of resource utilization, nutrition structure, green, and sustainability.

### 2.2. Review of Food Security Evaluation Research

Based on the internationally recognized basic definition of food security, international organizations, national institutions, and researchers have carried out evaluation and measurement of food security levels in different countries or regions. The main three ways to evaluate the performance of food security are single-index evaluation, household survey, and multi-index evaluation.

### 2.2.1. Single Index Evaluation

Single index evaluation refers to the evaluation of food security levels through a core index. For example, FAO uses "malnutrition rate" to evaluate food security. If the incidence of malnutrition reaches or exceeds 15%, the country is a country with food insecurity [8]. The World Food Conference proposed to use "grain carry forward inventory (ending inventory)" to measure food security and considered that the grain carry forward inventory (ending inventory) accounted for at least 17% of the food consumption of the year, which was the country with food security.

China is accustomed to taking per capita grain production and grain self-sufficiency rate as food security indicators: Chen Shaochong, 2009 [9] believed that in 2020, China's per capita annual share of grain needed to reach 420 kg to achieve food security. Zhu Ze (1997) and Ke Bingsheng (2007) [10,11] believed that a self-sufficiency rate of more than 85% could ensure food security. Lan Haitao (2007) and Jiang Changyun (2014) [12,13] believed that the grain self-sufficiency rate can better reflect China's food security situation and is internationally comparable. Drawing on the experience of Japan and South Korea, China will ensure that the self-sufficiency rate of wheat and rice is not less than 95% and that of corn is not less than 90% by 2020, which can basically ensure food security.

### 2.2.2. Household Survey

A household survey is a method of assessing regional food security levels by means of a microsampling survey. The Food and Nutrition Technical Assistance Program (FNTAP) calculates the consumption of 12 food groups in each family in the past 24 h through a household survey and obtains the family dietary diversity score (HDDS) [14,15]. The United Nations World Food Programme (WFP) calculates the food consumption score (FCS) based on dietary diversity, food consumption frequency, and the nutritional importance of food groups. The size of the score reflects the level of family food security [16,17]. The US government developed the Households Food Security Survey Module (HFSSM) at

the beginning of 1994 to monitor and assess the food security status of US households from three dimensions: families, adults, and children (Frongillo and Wolfe., 2010 [18], Weiser et al., 2015 [19], Heberlein et al., 2016 [20]). There are fewer studies in China in this area.

### 2.2.3. Multi-Index Evaluation

The multi-index evaluation method is the most widely used method in the theoretical circle to evaluate the level of food security. The multi-index evaluation method starts from the definition of food security, selects the variables with the highest relevance to food security from different dimensions, and then obtains the level of food security by statistical or mathematical methods.

The International Food Policy Research Institute (IFPRI) uses multiple indicators to measure the global hunger index (GHI). The more serious the hunger, the lower the food security [21]. The Economist Intelligence Unit (EIU) has designed the Global Food Security Index (GFSI) using 28 indicators of food supply, availability, quality, and food safety to measure the level of food security in different regions [22]. The Brazilian federal government has created a national food security and nutrition information system, including 50 indicators related to six aspects of food security. Maplecroft Venture Company mainly considers the sufficiency and stability of food supply and divides the results into five levels of "safety-extreme danger".

The central government of China implemented "The Provincial Governor Responsibility System for Food Security" in 2015 and formulated "The Assessment Method for the Provincial Governor Responsibility System for Food Security" to assess the food security work of each province. Academics often use comprehensive index systems to evaluate food security in China. First, evaluating China's food security level from the dimensions of productivity, accessibility, stability, and utilization d [23,24]. The second is to assess the level of food security based on different dimensions of the food supply and demand balance system [25,26]. The third is to assess the level of food security based on food production capacity or production increase potential [27,28]. Some scholars have also evaluated and measured from the perspectives of sustainable development [29,30], food security efficiency [31], and the "Risk Response" capacity of the food system [32,33].

In terms of calculation methods, they have basically adopted the two/three level weighting method. The weight setting methods include the expert consultation method, entropy index method, etc. The current multi-indicator evaluation research has basically reached a consensus on four major categories of indicators, including grain output, grain self-sufficiency rate, grain reserve capacity, and residents' nutrition and health.

### 2.3. Literature Review

To sum up, for the single index evaluation, it is obvious that there are too few indicators in the research, and the explanation is not comprehensive enough. The household survey method often attaches too much importance to the micro effect and ignores that food security is an important part of the macro attribute—national security. Therefore, the multi-indicator evaluation method is widely recognized by the theoretical community. At present, although the multi-indicator system can better reflect the situation of food security, there are three major shortcomings on the whole: ① focusing on the macro total amount, ignoring regional differences, ② focusing on the production of main production areas, ignoring the supply of non-main production areas, and ③ focusing on the quantity of food, ignoring the diary structure. These shortcomings are exactly what the research institute aims to optimize (Table 1).

In this paper, we first reduce the research dimension from the national level to the provincial level and observe the food security performance of different provinces. Secondly, the improved entropy weight method is combined with the analytic hierarchy process to balance the differences in regional food supply logic. Third, in the process of calculating

indicators, this study considered the differences in dietary habits and the rationality of dietary structure in different regions.

**Table 1.** Existing research defects and improved ways.

| Existing Research Defects | Improved Ways |
| --- | --- |
| Focusing on the macro, ignoring regional differences | Reducing the research dimension to the provincial level and observing the performance of different provinces |
| Focusing on the main production areas, ignoring the non-main production areas | Balance the differences in regional food supply logic, main/non-main production areas are, respectively, empowered |
| Focusing on the quantity of food, ignoring the diary structure | Considered the differences in dietary habits and the rationality of dietary structure in different regions |

### 3. Theoretical Analysis and Construction of Evaluation System

*3.1. Principles, Ideas, and Framework*

This paper strictly adheres to the internationally recognized definition of food security, takes the new connotation given by the "Great Food View" as guidance, and follows the principles of systematic, complete, scientific, and feasible to build a food security evaluation system to measure the food security of China's provincial administrative regions from 2009 to 2021.

There is a clear division of agricultural resources in China. According to the provincial grain production capacity, the traditional division of agricultural resources divides China's provinces into main grain production provinces and non-main grain production provinces. "*The Assessment Method for the Provincial Governor Responsibility System for Food Security*", which was formulated by the central government of China also divides China's provinces into main grain production provinces and non-main grain production provinces for separate assessment. However, food security under the "Great Food View" emphasizes the diversity and balance of the supply of food, which is not only the guarantee of the supply of grains but also the guarantee of the supply of staple and non-staple food, which means that the traditional agricultural resources division—depending on the grain production capacity—is no longer applicable to the food production division under the "Great Food View".

Therefore, in this paper, the division of the main food production provinces and non-main food production provinces is improved. We use the self-sufficiency rate of caloric intake to measure the food production capacity (see Section 4.1 for the calculation method). The main food production areas (which include several main food production provinces) and non-main food production areas (which include several non-main food production provinces) are, respectively, empowered, just like "*The Assessment Method for the Provincial Governor Responsibility System for Food Security*" do.

*3.2. Index System Composition and Index Selection*

According to the authoritative definition of food security, the core goal of food security is divided into three parts (Figure 1). First, quantitative goal: all people can obtain enough food at any time. Second, quality goal: the food obtained is safe, nutritious, meets dietary needs and food preferences, and the whole supply process is sustainable. The third is the stability and reliability of the above two. Regional food security is based on supply, and it is achieved through the market and the government jointly promoting the stable operation of the food supply system [34].

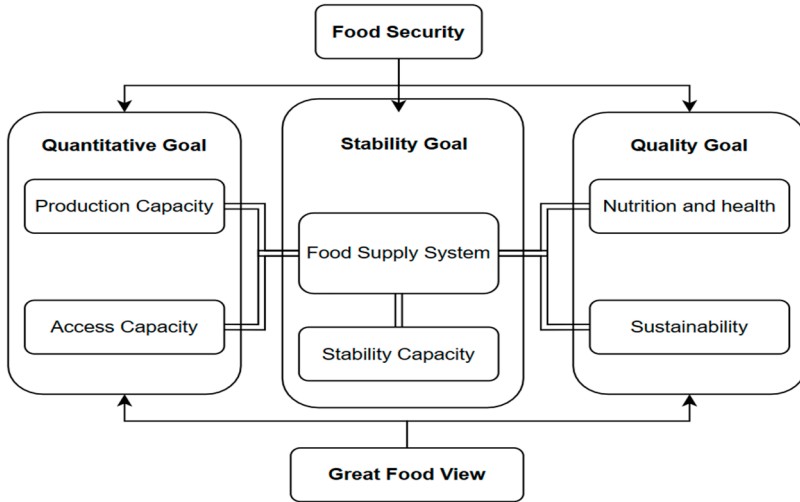

**Figure 1.** Analysis framework.

3.2.1. Meeting the Quantitative Goal Is the Quantitative Requirement for the Food Supply System

Although different provinces have different modes to access food, it is nothing more than internal production and external access. The capacity of each province to meet the quantitative goal of food security is evaluated from the two dimensions of production capacity (I1) and acquisition capacity (I2).

Production Capacity (I1)

Food production capacity is an important standard to measure food security [35]. This paper—referring to the relevant literature—uses the "C-D" production function to measure regional food production capacity from the output level (I11) and input level (I22) of the means of production. The level of output self-sufficiency rate represents the actual capacity of nutritional production of different regions. The food security concept under the "Great Food View" not only emphasizes the need to eat enough but also emphasizes the need to eat well. In this paper, the self-sufficiency rate of calories is used to evaluate the regional "eating enough" security capacity and the self-sufficiency rate of internationally recognized nutrients (protein and fat) is used to evaluate the regional "eating well" security capacity. The input level of means of production largely represents the local resource endowments and the government's attention to food security. The per capita (rural population) government investment in supporting agriculture agricultural expenditure, per capita farmland area, agriculture-related fixed assets per capita, and other indicators are selected to describe it.

Acquisition Capacity (I2)

It is difficult to rely solely on regional self-sufficiency to achieve regional food security with the present multi-category "staple" and "secondary" food demand [25]. The main food production areas produce crops with comparative advantages and obtain non-dominant crops through circulation means. Food safety in non-main production areas has gradually shifted from production safety to supply chain safety [33,34]. From the perspective of synergetics, this paper evaluates the regional food acquisition capacity from the perspectives of synergetic elements (I21) and synergetic abilities (I22). The indicators involved in synergetic elements are the per capita forest, grass area, and the calorie spillover degree of neighboring provinces. The synergetic abilities involve the total regional GDP, per capita disposable income, per capita output value of food processing, manufacturing, etc.

### 3.2.2. Meeting the Stability Goal Is the Stability Requirement for the Food Supply System

Internal and external systemic risks pose a great challenge to the anti-risk elasticity of the food supply system. The performance of food security under the "Great Food View" depends on the stability of the food supply (I3), which includes food quantity, price stability, and emergency response capability.

Stability Capacity (I3)

The performance of food security depends on whether the food supply is stable or not, and stable food supply includes normal supply and abnormal supply in response to emergencies. Under normal conditions, the stability of food price (price stability I31) and quantity (quantity stability I32) is the key standard of supply stability. Two price indexes are used to measure the price stability of the food supply system. The quantity fluctuation mainly comes from the production fluctuation and reserve investment, so the fluctuation rate of calorie production and per capita grain reserve expenditure are selected to measure the quantity stability. Under abnormal conditions, infrastructure construction is the material basis for facing emergencies. Therefore, the level of road network connectivity, per capita food emergency processing capacity, the density of food enterprises, and the number of employees in the storage industry are taken as specific indicators to evaluate the stability of food supply under abnormal conditions (emergency support capability I33).

### 3.2.3. Meeting the Quality Goal Is the Quality Requirement for the Food Supply System

Food supply is first oriented to meeting the nutritional and healthy food needs of regional consumers. Secondly, food security should be sustainable security based on ecological, resource, and economic sustainability. Therefore, the ability of provinces to meet the quality goal of food security is evaluated from the two dimensions of nutrition and health (I4) and sustainability (I5).

Nutrition and Health (I4)

The nutrition level (I41) and the reasonable degree of dietary structure (dietary rationality I42) are the important contents of food security under the "Great Food View". In this paper, the internationally recognized nutrition level indicator [35]—the proportion of animal-derived protein in total food protein intake, the proportion of vegetables and fruits—is selected to measure the nutritional level (I41) of residents. In terms of dietary structure (I42), there are huge differences in food dietary structure in different regions of China due to dietary habits, climate, and environment [36], according to the best dietary pattern given in the Chinese nutrition report, this research takes the deviation between the dietary habits of different regions and the best dietary pattern recommended by the state as an indicator to measure the health of regional dietary habits.

Sustainability (I5)

Green and sustainability are the eternal backgrounds of food security under the "Great Food View". It is particularly important to grasp the relationship between "production", "ecology" and "life". The first is to promote the agricultural production mode matching the resource and environment carrying capacity to ensure the sustainability of agricultural production [37]. The second is to coordinate "mountains, rivers, forests, farmlands, lakes, grass and sand" under the guidance of "The Bottom Line Thinking of Xi" and utilize natural resources in a diversified way. The third is to ensure that food producers and operators can obtain the necessary profit margins to ensure the sustainability of the agricultural economy. Only by ensuring the self-sustainability and self-development of food production resources and production systems can China achieve long-term and stable food security. Therefore, this paper evaluates sustainability (I5) from three dimensions of production sustainability (resource utilization I51), environmentally friendly (I52), and economic sustainability (I53).

According to the above ideas, the index system is constructed from five aspects: production capacity (I1), acquisition capacity (I2), stability capacity (I3), nutrition and

health (I4), sustainability (I5) and to measure the level of provinces food security and the ability to meet the three main goals. The specific indicators are shown in Table 2.

**Table 2.** Weight and weight difference between main and non-main food production areas.

| 1st Grade Indicators | 2nd Grade Indicators | 3rd Grade Indicators | Main-Prod Areas | Rank | Non-Main-Prod Areas | Rank | Difference | Rank of Diff |
|---|---|---|---|---|---|---|---|---|
| Production Capacity I1 | Output Level I1 | Calories Self-Sufficiency Rate | 0.0494 | 6 | 0.0170 | 27 | 0.0324 | 7 |
| | | Protein Self-Sufficiency Rate | 0.0451 | 7 | 0.0155 | 29 | 0.0296 | 9 |
| | | Fat Self-Sufficiency Rate | 0.0529 | 4 | 0.0182 | 25 | 0.0347 | 5 |
| | Input Level I12 | Per Capita Farmland Areas | 0.0565 | 2 | 0.0194 | 22 | 0.0371 | 3 |
| | | Per Capita Government Investment in Supporting Agriculture | 0.0785 | 1 | 0.0270 | 14 | 0.0516 | 1 |
| | | Agriculture-related fixed assets per capita | 0.0543 | 3 | 0.0187 | 24 | 0.0356 | 4 |
| Acquisition Capacity I2 | Synergy Elements I21 | Per Capita Forest And Grassland Area | 0.0252 | 19 | 0.0441 | 7 | 0.0189 | 11 |
| | | Calories Spillover Degree of Neighboring Provinces | 0.0309 | 15 | 0.0541 | 5 | 0.0232 | 10 |
| | Synergy Ability I22 | Regional GDP | 0.0434 | 8 | 0.0760 | 2 | 0.0326 | 6 |
| | | Per Capita Output Value of Food Processing and Manufacturing Industry | 0.0414 | 10 | 0.0726 | 3 | 0.0311 | 8 |
| | | Per Capita Disposable Income | 0.0519 | 5 | 0.0909 | 1 | 0.0390 | 2 |
| Stability Capacity I3 | Price Stability I31 | Fluctuation of Rural Retail Price Index | 0.0169 | 25 | 0.0211 | 20 | 0.0043 | 20 |
| | | Fluctuation of Urban Retail Price Index | 0.0165 | 26 | 0.0207 | 21 | 0.0042 | 21 |
| | Quantity Stability I32 | Grains Reserve Expenditures Per Capita | 0.0433 | 9 | 0.0542 | 4 | 0.0109 | 12 |
| | | Fluctuation Rate of Calories Production | 0.0249 | 20 | 0.0311 | 11 | 0.0063 | 15 |
| | Emergency Support Capability I33 | Density of Food Enterprise | 0.0317 | 14 | 0.0397 | 8 | 0.0080 | 14 |
| | | Number of Employees in Storage Industry (per 10k people) | 0.0398 | 11 | 0.0499 | 6 | 0.0100 | 13 |
| | | Highway Length Per Capita | 0.0196 | 24 | 0.0245 | 18 | 0.0049 | 18 |
| | | Per Capita Food Emergency Processing Capacity | 0.0236 | 21 | 0.0296 | 12 | 0.0060 | 17 |
| Nutrition And Health I4 | Nutritional Level I41 | Proportion of Vegetables and Fruits | 0.0274 | 16 | 0.0267 | 15 | 0.0007 | 28 |
| | | Proportion of Animal Protein | 0.0384 | 12 | 0.0373 | 10 | 0.0010 | 27 |
| | Dietary Rationality I42 | Deviations between The Dietary Structure and Recommended Values of Meat and Grains | 0.0269 | 18 | 0.0261 | 17 | 0.0007 | 30 |
| | | Deviations between The Dietary Structure and Recommended Values of Vegetables and Grains | 0.0272 | 17 | 0.0264 | 16 | 0.0007 | 29 |
| Sustainability I5 | Resource Utilization I51 | Proportion of Rural Green Power Generation in Agricultural Power Generation | 0.0235 | 22 | 0.0279 | 13 | 0.0044 | 19 |
| | | Per Capita Output of Forest and Grassland Products | 0.0207 | 23 | 0.0245 | 19 | 0.0038 | 22 |
| | Environmentally Friendly I52 | Unit Usage of Pesticides and Fertilizers | 0.0148 | 28 | 0.0176 | 26 | 0.0027 | 24 |
| | | Agricultural Carbon Emissions Intensity [38] | 0.0331 | 13 | 0.0392 | 9 | 0.0061 | 16 |
| | Economic Sustainability I53 | Price Index of Means of Agricultural Production | 0.0125 | 30 | 0.0148 | 30 | 0.0023 | 26 |
| | | Disparity Between Urban and Rural Disposable Income | 0.0135 | 29 | 0.0160 | 28 | 0.0025 | 25 |
| | | PPI of Agriculture Products | 0.0164 | 27 | 0.0194 | 23 | 0.0030 | 23 |

## 4. Data and Methods

This research combines the AHP with the improved entropy weight method to empower the food security evaluation system constructed above, so as to further conduct an empirical evaluation of China's provincial food security situation.

(1) The first level index weight is determined through the AHP. (2) The second and third level index weights are determined firstly by the improved entropy weight method, and then the second and third level index weights are subsequently revised by AHP.

*4.1. Research Method Selection*

4.1.1. Analytical Hierarchy Process

Analytic hierarchy process (AHP) is widely used for systematic analysis of problems in social, economic, and management fields. It combines qualitative and quantitative analysis and is suitable for decision-making problems with hierarchical and staggered evaluation indicators. In this research, there are differences in natural endowments and comparative advantages between the main and the non-main food production areas in China, and the weights of food security indicators are inconsistent [16,39]. Previous research underestimated the level of food security in non-main food production areas due to weak food production capacity. Therefore, this paper uses the AHP to empower the 1st grade indicator of the main/non-main food production areas, respectively, reflecting the differences in food security approaches between the main production areas and non-main production areas of the importance of the first-level indicators.

4.1.2. Improved Entropy Weight Method

The entropy weight method is a multi-index comprehensive evaluation method that objectively determines the index weight according to the different information entropy and variation degree of the index. It is widely used in the research of multi-index evaluation in ecology, management, economics, and other fields. Although there are differences in the ways of food security among regions, the underlying logic of ensuring food security is consistent. After selecting the entropy weight method to initially determine the weights of the 3rd grade of indicators, the weights of the second and third levels of indicators are corrected by the AHP, and the entropy weight method is improved with reference to the existing literature. The improved entropy evaluation model is as follows:

Step 1: Standardization, in order to eliminate the dimensional differences of different indicators.

Positive and negative indicators preliminary standardization method:

$$y_{ij} = x_{ij} - x_{imin}/x_{imax} - x_{imin}, \ y_{ij} = x_{imax} - x_{ij}/x_{imax} - x_{imin} \tag{1}$$

Since the panel data has both regional and time attributes, it is necessary to consider the impact of two aspects in the standardization process. Note: $Z_{nk}$ is the standardization matrix of the two attributes of region and index, and $Z_{tk}$ is the standardization matrix of the two attributes of time and index. The final standardized matrix is: (improvement process)

$$Z_{ntk} = \sqrt{Z_{nk} \times Z_{tk}} \tag{2}$$

Step 2: Determine the index feature weight. After standardization, the numerical contribution of the *t* year in the *n* region is obtained.

$$Y_{ntk} = Z_{ntk}/\sum_{n=1}^{N}\sum_{t=1}^{T} Z_{ntk} \tag{3}$$

Step 3: Determine the index information entropy $E$ and information utility value $d$. The calculation methods are:

$$E_k = -\frac{1}{Ln(NT)}\sum_{n=1}^{N}\sum_{t=1}^{T} Y_{ntk} * Ln(Y_{ntk}), d_k = 1 - E_k \tag{4}$$

Step 4: Determine the weight. The weight of the index *k* is:

$$W_k = d_k/\sum_{k=1}^{K} d_k \tag{5}$$

4.1.3. Index Safety Score Conversion Method

In order to obtain a more intuitive food security score, this paper draws on the methods of Gu Haibing and Zhang Anjun [40] and adjusts them. The risk value of the positive index $(S_L)$ is assigned to 0, and the risk value of the negative index $(S_M)$ is set with reference to

the relevant research of the *Rural Economic Survey Division* [17], Changyun Jiang [41] and others. The conversion method of positive and negative index scores is as follows:

$$S_{+} = [1 - {}^{X - X_L}/{}_{X_L - X_M} \times (S_M - S_L)] \times 100 \tag{6}$$

$$S_{-} = [{}^{X - X_L}/{}_{X_L - X_M} \times (S_M - S_L)] \times 100 \tag{7}$$

*4.2. Data Sources and Index Calculation*

The data in this research are all from relevant *Statistical Yearbooks*, which are collected by China's County, Municipal and Provincial Statistical Bureaus, and released by the National Bureau of Statistics every year. It is the most authoritative statistical data officially released by China. See Table 3 below for details.

**Table 3.** Data sources.

| Indicators of Evaluation System | Data Sources |
| --- | --- |
| Per Capita Farmland Areas, Per Capita Forest And Grassland Area, Per Capita Government Investment in Supporting Agriculture, Agriculture-related fixed assets per capita | *China Rural Statistical Yearbook* |
| Regional GDP, Per Capita Disposable Income, Number of Employees in Storage Industry (per 10k people), Highway Length Per Capita | *China Statistical Yearbook* |
| Per Capita Output Value of Food Processing and Manufacturing Industry, Density of Food Enterprise | *China Industrial Statistical Yearbook* |
| Price Index of Means of Agricultural Production | *Yearbook of China's Urban (Town) Life and Price* |
| Per Capita Food Emergency Processing Capacity | *Provincial Five-Year Plans* |
| Fluctuation of Rural Retail Price Index, Fluctuation of Urban Retail Price Index, PPI of Agriculture Products | *Yearbook of China's Urban (Town) Life and Price* and calculate by authors |
| Calories Self-Sufficiency Rate, Fat Self-Sufficiency Rate, Protein Self-Sufficiency Rate, Calories Spillover Degree of Neighboring Provinces, Fluctuation Rate of Calories Production, Proportion of Vegetables and Fruits, Proportion of Animal Protein, Deviations between The Dietary Structure and Recommended Values of Meat and Grains, Deviations between The Dietary Structure and Recommended Values of Vegetables and Grains | *China Rural Statistical Yearbook, China Statistical Yearbook* and calculate by authors |
| Grains Reserve Expenditures Per Capita, Disparity Between Ur-ban and Rural Disposable Income | *China Statistical Yearbook* and calculate by authors |
| Proportion of Rural Green Power Generation in Agricultural Power Generation, Per Capita Output of Forest and Grassland Products, Unit Usage of Pesticides and Fertilizers | *China Rural Statistical Yearbook* and calculate by authors |

The calculation method of self-sufficiency rate of calories, protein, and fat is: [(Provincial food production) $\times$ (Corresponding nutrition index)]/[(Per capita food consumption of provincial households) $\times$ (Provincial population $P_{pi}$) $\times$ (Corresponding index of nutrition)]. Food types of output and consumption include rations, meat (pigs, poultry, cattle, sheep), vegetables and edible mushrooms, milk, eggs, aquatic products, and fruits ($x_i$). The corresponding nutrition index of food is based on the food nutrition content in the Table of Chinese Food Composition ($y_i$) calculated ($\sum_i xy$).

Provinces are divided into main food production provinces and non-main food production provinces in the following ways: (1) According to the "*The Assessment Method for the Provincial Governor Responsibility System for Food Security*", the main grain production areas and non-main grain production areas are obtained. (2) Calculate the average food caloric self-sufficiency rate ($\mu$) and the standard deviation of caloric self-sufficiency rate ($\sigma$) in the main grain production areas. (3) According to $\Omega = \mu - Z_{\frac{\alpha}{2}} \frac{\sigma}{\sqrt{n}}$ the lower confidence

limit of 95% confidence interval in the main food production areas can be determined. (4) The provincial-level divisions where the caloric self-sufficiency rate is higher than $\Omega$ are classified as the main food production areas and less than $\Omega$ are classified as the non-main food production areas. The main food production areas are: Hebei, Inner Mongolia, Liaoning, Jilin, Heilongjiang, Jiangsu, Anhui, Shandong, Henan, Hubei, Hunan, Sichuan, Jiangxi, Tibet, Ningxia, and Xinjiang. The non-main food production areas are: Beijing, Tianjin, Shanghai, Chongqing, Qing, Fujian, Guangdong, Hainan, Guizhou, Shaanxi, Zhejiang, Gansu, Yunnan, Shanxi, and Guangxi.

The calculation method of calorie spillover of neighboring provinces ($R_i$) is as follows: (1) Obtain the calorie spillover amount (calories production) of the provinces that are geographically adjacent to the province $R_{pj}$ − calories consumption $R_{cj}$). (2) By $R_i = \sum R_{pj} - R_{cj}/R_{cj}$ obtain the spillover amount.

The fluctuation rate of calorie production is calculated using the coefficient of variation of the deviation between the actual value and the trend value in the past three years [42]. (1) Use ARIMA to fit the trend. (2) Calculate the moving average of absolute deviation in the past 3 years $ma\_d_i = \sum_{j=0}^{2}|d_{i-j}|/3$. (3) Calculate the standard deviation of absolute deviation in the past 3 years.

The calculation method of deviation between meat and grain dietary structure and recommended value ($\mu_i$) is as follows: (1) According to the per capita food consumption of household residents in different regions, the meat and grain dietary habits (proportion) of residents in different regions can be obtained ($\rho_i$). (2) Recommended proportion of dietary structure according to the "*Chinese Residents' Balanced Diet Pagoda*"$\Delta$. (3) According to $\mu_i = |\rho_i - \Delta|$ calculated.

Grading method of food security: use mathematical statistics $3\sigma$ principle for reference [43], select 1–2 times the standard deviation of the mean value of plus or minus food security scores to determine the critical point of food security evaluation division under the "Great Food View" in different regions, and establish regional levels.

## 5. Food Security Evaluation

In this study, 22 experts were invited to participate in the weight consultation of the 1st grade indicators by using the analytic hierarchy process. The questionnaire preparation group took into account the academic professionalism and practical experience of the experts. A total of 50% of the experts came from the provinces in the main food production areas and 50% from the provinces in the non-main food production areas. Among them, 15 were members of the registered expert group of the local food and material reserve bureau and the emergency bureau, 7 were senior managers of large food (grain and oil) processing enterprises, 68.18% of the experts have doctoral degrees, and the executives had more than 20 years of experience in the food market. The C.R. value of the consultation result was less than 0.1, and the judgment matrix had satisfactory consistency. Figure 2 is the weight of the five subsystems (1st grade indicators) of food security in main food production areas and non-main food production areas.

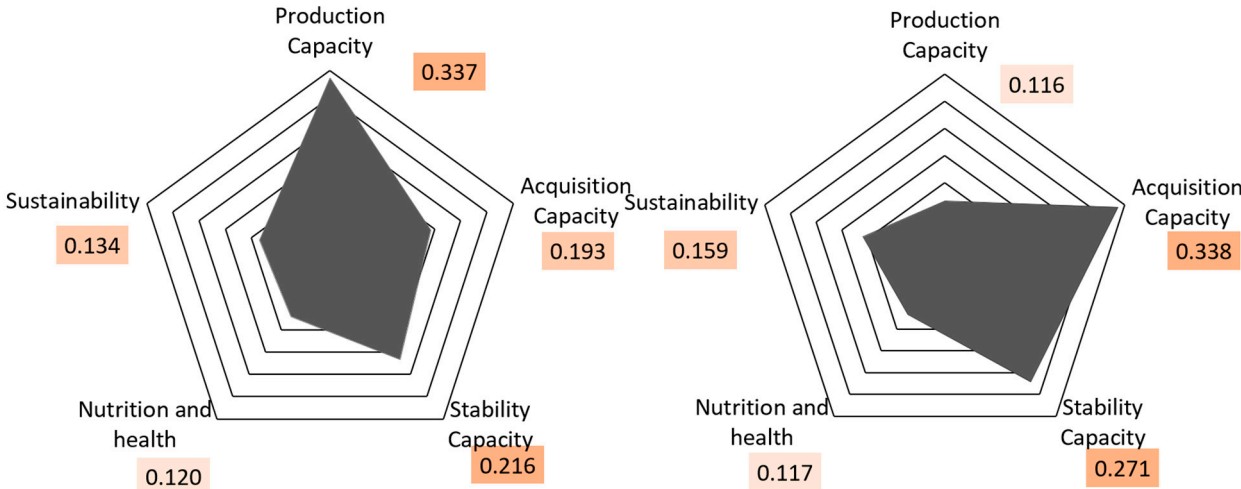

**Figure 2.** The weight of 1st grade indicators in main (**left**) and non-main (**right**) food production areas.

*5.1. Differences in Food Security Structure between Main and Non-Main Food Production Areas*

From the first-level indicators (Figure 2), for the main food production areas, production capacity I1 accounted for the largest weight (0.337), followed by stability capacity I3 (0.216), acquisition capacity I2 (0.193), sustainability I5 (0.134) and nutrition and health I4 (0.120). This conclusion is similar to the conclusion of the macro measurement of China's food security [15]. For non-main production areas, acquisition capacity I2 (0.338) is the most important to ensure food security, followed by stability capacity I3 (0.271), sustainability I5 (0.159), nutrition and health I4 (0.117) and production capacity I1 (0.116). Therefore, it can be concluded that there is a strong heterogeneity in the way of ensuring food security in the main production areas and non-main production areas, which further confirms the rationality of the respective weighting of the first-level indicators in the main food production areas and non-main food production areas, green and sustainable food supply I5 and nutrition and health. However, the sustainability I5 of food supply and the nutrition and health I4 does not account for a large proportion of the food security rating system, which also shows that China's food security still pays attention to the performance of "quantity" but does not pay enough attention to "quality". Zhu Jing and others reached a similar conclusion through China's grain production reserves and price data.

From the perspective of the core goal of food security, meeting the food quantitative goal (the sum of the weights of production capacity and acquisition capacity I1 + I2) is the most important starting point for ensuring food security. It occupies the highest weight in both the main production areas (0.5295) and the non-main production areas (0.4533). The quality goal of food security (I4 + I5) (0.254) in the main food production areas is more signification than the stability goal (I3) (0.216), while the two in the non-main food production areas are basically the same.

From the perspective of 2nd grade indicators (Table 4), in the main food production areas, the top three 2nd-level indicators of the importance of food security are input level I12 (0.19), output level I11 (0.15), and the synergy ability I22 (0.14). On the one hand, the high weight of the input and output levels reflects that food production is particularly important to the food security of the main production areas and even the country. On the other hand, the high weight of synergy ability shows that even the main food production areas with high self-sufficiency rates should have sufficient capacity to obtain food with multiple synergies under the "Great Food View". In non-main food production areas, the second-level indicators of the top three importance levels of food security are the synergy ability I22 (0.25), emergency support capability I33 (0.13), and quantity stability I32 (0.10). The average self-sufficiency rate of calories in non-main production areas is less than 70%, and that in Beijing, Shanghai, and other municipalities is less than 15%. Whether food

can be supplied in a stable and sufficient way is the most important standard for food security in non-main production areas. The "Synergy Capability" indicator measures the ability to "provide sufficient", while the two main secondary indicators, emergency support capability and quantitative stability, are important parts of measuring the "stability of supply" in all cases.

**Table 4.** The 2nd grade indicators' weights of main and non-main food production areas.

|  | **Most Important** | **Second Important** | **Third Important** |
|---|---|---|---|
| Main prod | Input Level I11 (0.19) | Output Level I11 (0.15) | Synergy Ability I22 (0.14) |
| Non-main prod | Synergy Ability I22 (0.25) | Emergency Support Capacity I33 (0.13) | Quantity Stability I32 (0.10) |

From the 3rd grade indicators (Table 2), the weight convergence between the main food production areas and non-main food production areas focuses on the nutrition level and dietary rationality-related indicators, which indicates that although there are objective differences between the main food production areas and non-main food production areas in terms of food access, the emphasis on dietary structure and nutrition level is basically the same, and this research conclusion is consistent with the actual performance.

*5.2. Provincial Food Security Evaluation*

According to the food security performance of 31 provinces in China from 2009 to 2021 under the "Great Food View" (Figure 3). First, the level of food security in all provinces entered the ranks of "safe" food security rating in 2020. Second, the food security level of the four main municipalities remains high, and high food security vulnerability areas [21] (such as Shanghai, Beijing, Ningxia, Tianjin, etc.), have reached a high level of food security, indicating that the mismatch between "high vulnerability and low security" has been greatly alleviated. Third, the five provinces with the lowest food security level in 2020 (Hainan, Qinghai, Shanxi, Jiangxi, and Guizhou) not only had a low ranking for a long time but also a growth rate of scores lower than the national average. The problem of "low growth rate, difficult growth" has emerged. This phenomenon shows that China's food security level has entered a key link under the "Great Food View". In the face of the new requirements of the transformation of the agricultural and food structure for food security under the Great Food View, China's food security level continues to improve into a key link.

Table 5 shows the ranking of the ability to meet the three core goals of food safety under the "Great Food View".

**Table 5.** Ranking of ability to meet the three coal goals of food security under the "Great Food View".

|  | **Top Five (Left to Right, 1–5)** | **Last Five (Left to Right, 31–27)** |
|---|---|---|
| Quantitative Goal | Inner Mongolia, Jilin, Heilongjiang, Beijing, Shanghai | Hainan, Qinghai, Guangxi, Jiangxi, Guizhou |
| Stability Goal | Heilongjiang, Shanghai, Jilin, Inner Mongolia, Beijing | Sichuan, Jiangxi, Yunnan, Guizhou, Guangdong |
| Quality Goal | Hunan, Zhejiang, Jiangxi, Guangxi, Jiangsu | Shanxi, Tibet, Inner Mongolia, Liaoning, Beijing |

1. Quantitative goal of food security under the "Great Food View". The three main food-producing provinces (Inner Mongolia, Jilin, Heilongjiang) and two municipalities directly under the Central Government (Beijing, Shanghai) have the strongest ability to meet the quantitative goal of food security. For Inner Mongolia, Jilin, and Heilongjiang the average caloric self-sufficiency rate of the three provinces (calculated by 2021, the same as below) is 2.82 times the national average, the protein self-sufficiency rate is 2.59 times the national average, and the fat self-sufficiency rate is nearly 3 times the national average. Strong food production capacity lays a solid foundation for meeting the quantitative goal of food security under the "Great Food View". Although the rate of food self-sufficiency in Beijing and Shanghai is very low (less than 15%), they have excellent regional layouts and

strong economic strength. Their ranks of food synergy ability (I22) are among the top two in China. So, they have a diversified food supply system to meet the quantitative goal of food security under the Great Food View.

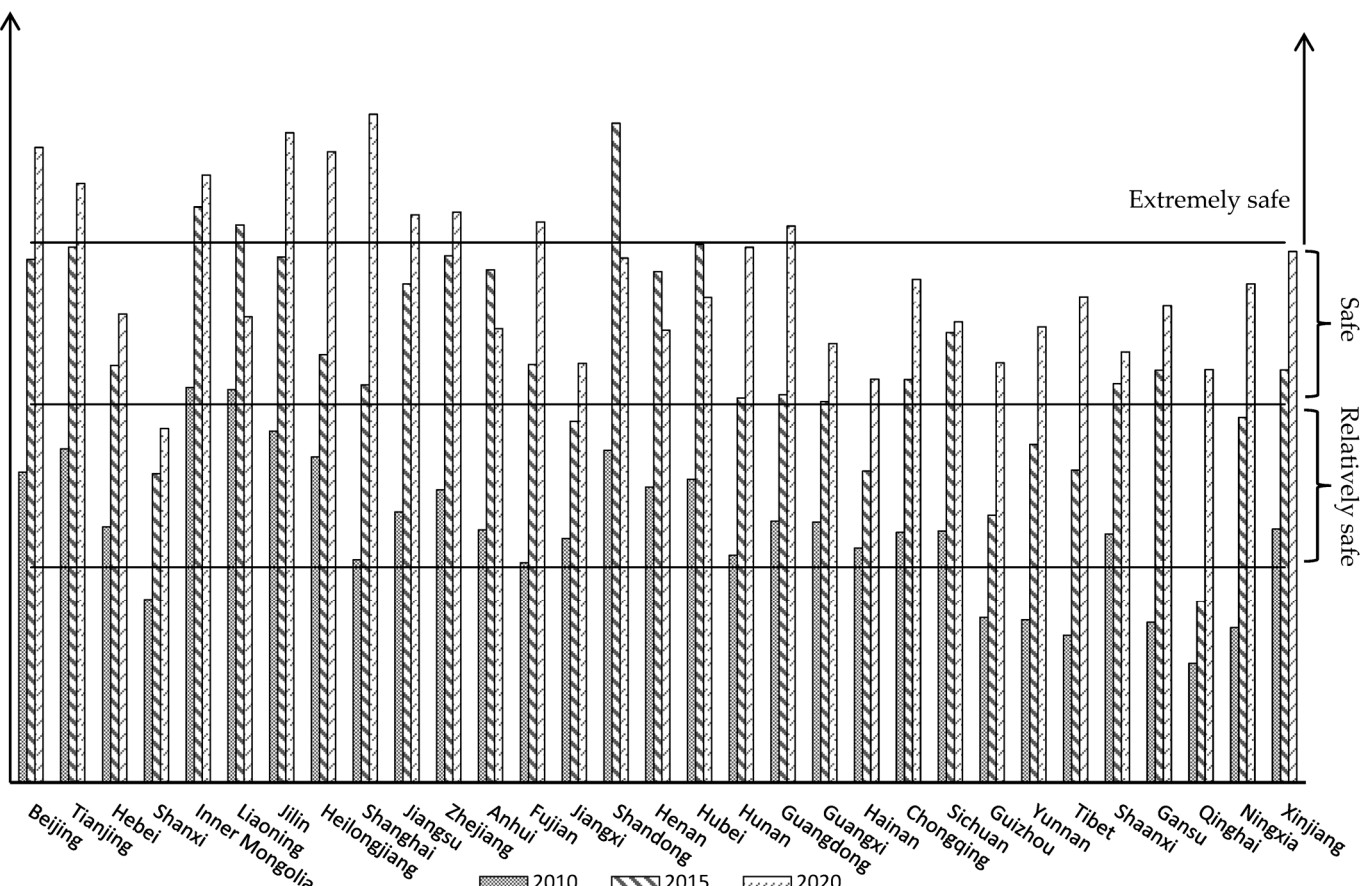

**Figure 3.** Performance of food security in provinces.

2. Stability Goal of food security under the "Great Food View". Heilongjiang, Shanghai, Jilin, Inner Mongolia, and Beijing still rank in the top five. The stable supply of food under abnormal conditions is an important manifestation of regional food security. At present, Beijing and Shanghai and other municipalities have established a food storage system with the coordination of government reserves and social responsibility reserves. The high density of food wholesale and retail enterprises provides objective support for the effectiveness of a strong storage system, forming the cornerstone to meet the goal of food security and stability. The fluctuation rate of calorie production in the main food-producing provinces in the north, such as Inner Mongolia, Jilin, and Heilongjiang, is only 68% of the national average. In addition, the main food-producing regions have natural advantages in maintaining the food supply and price stability, which ensure the extremely high stability of food supply. However, the key indicators for ensuring food supply stability in Sichuan, Jiangxi, Yunnan, Guizhou, Guangdong, and other places with the worst food supply stability are lower than the national average. The per capita grain reserve expenditure is 71.52% of the national average, and the per capita grain emergency processing capacity is 61.55% of the national average. The difficulty of emergency support caused by mountainous and hilly terrain is also an important reason for the low food security and stability indicators.

3. Quality goal of food security under the "Great Food View". There are relatively clear differences between the north and the south of China. The top five are Hunan, Zhejiang, Jiangxi, Guangdong, and Jiangsu, all located in the south of China, and the last five are Shanxi, Tibet, Inner Mongolia, Liaoning, and Beijing, all located in the north of China. First, in terms of the representative indicator of agricultural sustainability—agricultural carbon

emission intensity—the last five provinces are 11.75% higher than the national average level. Secondly, in terms of dietary habits, the *Scientific Research Report on Dietary Guidelines for Chinese Residents (2021)* pointed out that the diet in the middle and lower reaches of the Yangtze River, represented by Zhejiang and Shanghai, can be the representative of an Oriental healthy diet, and the overweight and obesity rates of adults in Hunan (22.3%), Zhejiang (24.3%), Jiangxi (20.1%), Guangxi (14.6%), and several provinces are lower than the national average (28.1%), while those in Shanxi (29.2%), Tibet (26.7%)Mongolia (37.7%), Liaoning (32.5) and Beijing (40.9%) are almost all significantly higher than the average rate. Therefore, reasonably revising the residents' dietary structure', especially in the northern regions is an important aspect of China's food security reaching high-quality development under the "Great Food View".

*5.3. Regional Food Security Performance*

5.3.1. Performance of Food Security in Main and Non-Main Food Production Areas under the "Great Food View"

The main food-producing areas in China shoulder the task of ensuring national food security. With 56% of the Chinese population, the main food production areas produce 78.6% of grain, 67.1% of meat, and 70.1% of milk. From the perspective of the overall food security level and the temporal change trend of the performance of the first-level indicators in Figure 4. The food security level of the main food production areas has improved year by year from 2009 to 2014 and gradually stabilized from 2014 to 2020. According to the analysis of the performance of the first-level indicators, acquisition capacity (I2), stability capacity (I3), and nutrition and health level (I4) were in a stable growth state from 2009 to 2020. However, the fluctuation trend of food production capacity (I1), which has the largest weight of food security in the main food production areas, firstly shows a rising and then stable fluctuation trend. This phenomenon shows that the improvement of diversified food supply and emergency security capacity in the main food production areas is an important driving force for the continuous rise of food security under the "Great Food View". However, it is worth noting that the sustainable (I5) level of the food security system in China's main food production areas has always been at a low level.

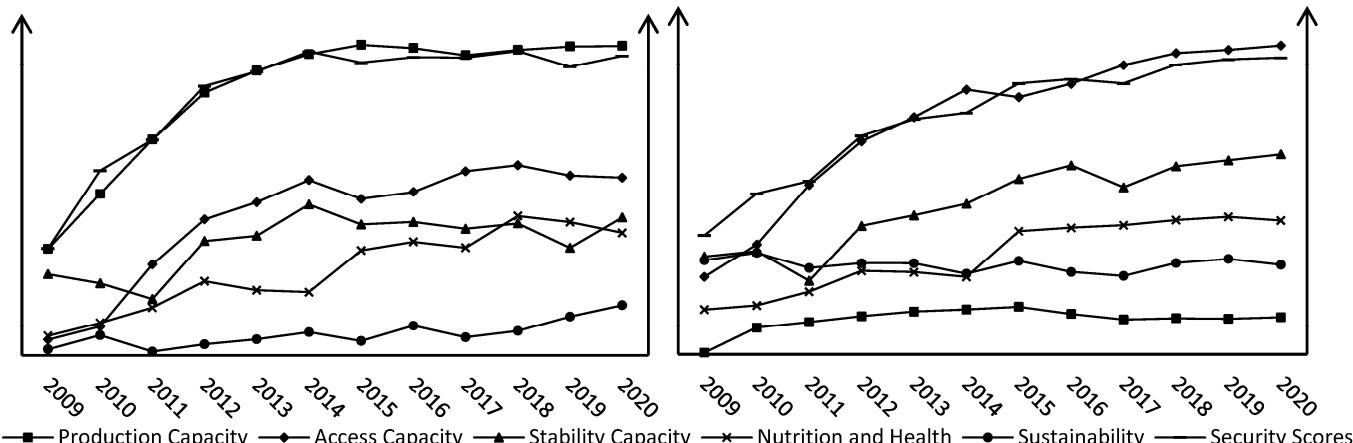

**Figure 4.** Performance of food security in main (**left**) and non-main (**right**) food production areas.

In the non-main food producing areas in China, from the perspective of the performance of food security and the change trend of the first-level indicators, the change trend of food security level in the non-main food production areas is similar to that in the main production areas, but the rate of increase is relatively slow, and after the food security level in the main food production areas began to stabilize in 2014, it continued to increase at an average annual rate of 0.84%. In terms of 1st-level indicators, the acquisition capacity I2 (0.338) and stability capacity (0.271), which are the top 2 indicators for the food security level of non-main production areas, increased at an annual average growth rate of 19.06%

and 7.14%, respectively, from 2009 to 2020, becoming an important engineer for the food security level of non-main production areas. Meanwhile, the food nutrition and health (I4) level of non-main production areas has steadily improved year by year. However, in the same main food production areas, the sustainability (I5) of food security has always been in a lower position and has not been highlighted, which probably indicates that although China's food security level continues to improve, it may belong to the extensive environmental growth.

5.3.2. Distribution Evolution of Food Security Level in Main and Non-Main Food Production Areas Based on Kernel Density Estimation

In order to further analyze the evolution characteristics of food security in main and non-main food production areas, this paper uses the Kernel density estimation method to investigate the regional differences in food security. The overall evolution trend of food security in main and non-main production areas is shown in Figure 5.

1. From the distribution position: The center axis of the Kernel density curve indicated the regional average performance. From Figure 5, we can see that the center axis of the Kernel density curve of food security in the main production areas and non-main production areas is both moving to the right, which proves that the overall level of food security in the main production areas and non-main production areas is increasing year by year, which is consistent with the previous conclusions.

2. From the distribution pattern: The sharper and higher peak of the Kernel density curve for food security indicates that the distribution of food security performance in the region is more concentrated, and the gap between them is smaller. The wider of Kernel density curve peak is, the more unconcentrated and bigger it is. In terms of distribution pattern, the Kernel density curve of food security in main and non-main food production areas is quite different. The distribution pattern of the Kernel density curve of food security in the main production areas shows the evolution characteristics from "thin peak" to "wide peak" and then to "thin peak", and the peak value first decreases and then increases, indicating that before 2017. From 2010 to 2017, although the overall food security in the main production areas was continuously improving, the internal differences in the performances of food security were also increasing, high food security provinces and low food security provinces coexist, and the gap is growing. However, after 2017, this phenomenon improved, which is reflected in the "thin peak" and high peak of the Kernel density curve in the main production areas. In non-main production areas, the distribution pattern of the Kernel density curve of food security shows the evolution characteristics from "thin peak" to "wide peak" and then to "more wider peak", and the peak value is constantly decreasing, which indicates that although the overall food security score of non-main production areas is constantly rising, it is unbalanced. For example, in 2021, the food security score range of provinces in non-main production areas will increase by 41.21% compared with that in 2010, and the standard deviation will increase by 54.50%. The gap within the region will continue to increase in the number of peaks.

3. From the number of peaks: In 2021, there is an obvious main peak and a side peak in the main food production areas, but the height of the side peak is lower than the main peak, indicating that the food security in the main production areas will begin to show a "Gradient effect", and there will be a small group with high food security performance. The "Gradient effect" is more obvious in non-main food production areas. There are main peaks and side peaks in each Kernel density curve in 2010, 2013, 2017, and 2021, which further verifies the unbalanced development of food security in non-main production areas.

4. Finally, in terms of distribution extension: The Kernel density curve of the main production areas in 2017 has a significant right tailing, which indicates that there are both high food security provinces and low food security provinces in the main production areas, which also validates the previous conclusions, but the tailing has

significantly reduced in 2021, and the development balance has significantly improved. However, the right tailing in non-main food production areas has become "thicker" and longer year by year, indicating that there are not only high food security provinces and low food security provinces, but also the growth rate of provinces with low food security scores is lower than that of provinces with high scores, leading to further widening the gap between high and low provinces.

5.  To sum up, the overall score of food security in both main and non-main food production areas is increasing from 2010–2021, but the development of food security in provinces within main production areas is more balanced than that in non-main production areas. The distribution shape, peak number, and distribution extensibility of the Kernel density curve all indicate that the "Matthew effect" of food security begins to appear in non-main production areas.

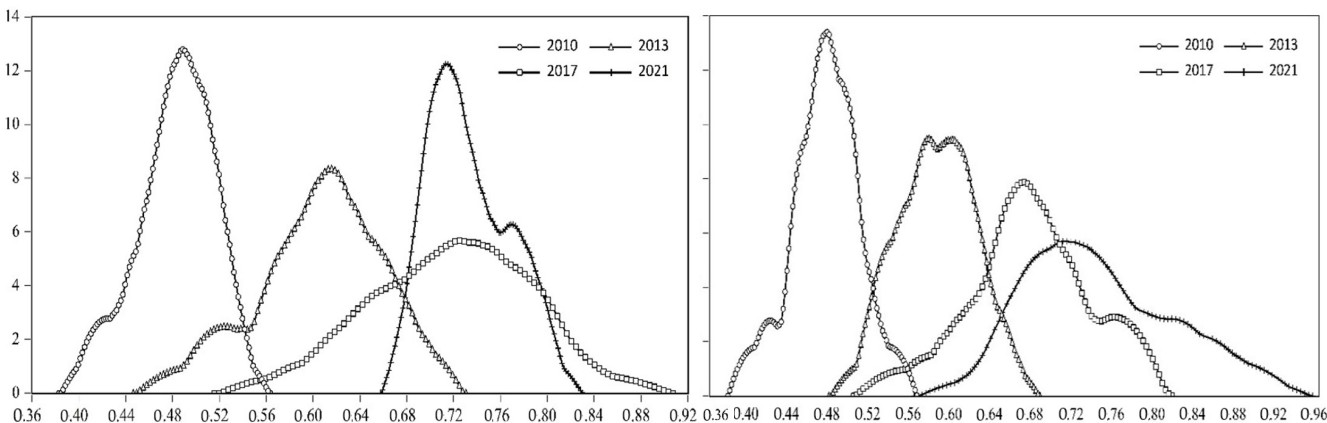

**Figure 5.** Kernel Density estimation of food security in main (**left**) and non-main (**right**) food production areas.

## 6. Discussion, Conclusions, and Enlightenment

### 6.1. Conclusions

Through the above analysis, this research constructs China's provincial food security evaluation system under the "Great Food View" and evaluates the current situation and trend of food security in China's provinces (except Hong Kong, Macao, and Taiwan) from 2009 to 2021 by using this system. The main conclusions are as follows: (1) There are obvious differences in the food security structure of China's main and non-main food-producing areas under the "Great Food View". Meeting the quantitative target of food security is the most important indicator, and strong production capacity and strong acquisition capacity can well meet the quantitative target of food security. (2) All provinces in China have entered the ranks of food security "security" and above under the "Great Food View". Areas with high food security vulnerability (such as Shanghai, Beijing, Ningxia, Tianjin, etc.), have reached a high level of food security, but the growth rate has begun to slow down. (3) The five provinces with the lowest food security level (Hainan, Qinghai, Shanxi, Jiangxi, Guizhou) have been in a low ranking for a long time, and their growth rate is also lower than the average level. (4) The sustainable degree of the food security system in China has always been at a low level, and the food security in non-main production areas has appeared an unbalanced and unequal "Matthew effect".

### 6.2. Discussion

Based on the "Great Food View", this paper constructs a food security evaluation system in China, innovatively combines subjective and objective empowerment methods, and conducts an empirical analysis of China's domestic food security.

The food security evaluation system constructed in this paper under the "Great Food View" has greatly optimized the existing food security evaluation system in China. First, it has completed the transformation of the evaluation object from "grain" to "food" [24,44],

which is closer to the forefront of international food security research [45,46]. The second is to creatively incorporate the differences in dietary habits and the rationality of dietary structure into the evaluation system, which is more consistent with the objective reality and the international definition of food security [FAO]. The third is to consider the logical differences in food supply between regions that have not been considered before [29]—whether food is mainly produced by oneself or purchased from outside—which is the most basic difference in the way of food security in different regions.

In the empirical analysis of China's domestic food security, this paper combines China's policy practice with theoretical research, not only exploring the transformation of China's agricultural resource zoning from "grain zoning" to "food zoning", but also digitizing the objective differences through AHP and entropy weight method. The research conclusion also reversely verified the scientific effectiveness of the separate weighting of the main and non-main food-producing areas, which largely avoided the error in the estimation of food security in different functional areas in previous studies. Because of this ingenious treatment, this study also draws some interesting new conclusions. For example, (1) although the food production capacity of non-main production areas in China is slowly declining, the level of food security is slowly increasing. Yet currently some literature generally believes that the level of food security in non-main production areas is declining [21]. This is because the main and non-main production areas are empowered together. The food security level of non-main production areas is underestimated due to their weak food production capacity. (2) The study has obtained low food security level regions (Hainan, Qinghai, Shanxi, Jiangxi, Guizhou), and similar conclusions have been obtained in other studies [47], but this paper further determines that these regions are not only in a low ranking for a long time but also in a lower growth rate than the average level. (3) This study uses the Kernel density estimation to find the "Matthew effect" of food security in non-main production areas, which is also confirmed by the Theil index in other studies [47], indicating that the inequality of food security in non-main production areas in China exists objectively. If we can further reveal the reasons behind it scientifically, it will be the direction of my future research.

*6.3. Enlightenment*

Based on the above conclusions, we can obtain some enlightenment: (1) Adhering to the concept of the "Great Food View", starting from a broader range of food, developing food resources in an all-round and multi-channel way, and better meeting the food quantity and diversified consumption needs. (2) Under WTO rules, China should increase support for agricultural subsidies and implement targeted and oriented assistance. According to local conditions, guide subsidies, and assistance to provide the building of small and medium-sized agricultural hydropower, the purchase of agricultural machinery [48], and the use of special breeding technology in the countryside, so as to ensure that the level of agricultural greening, modernization, and self-sufficiency in fat will be improved. (3) Quantify the quality of farmland resources, and establish more precise compensation and protection policies for farmland. (4) Rationally optimize the distribution of agricultural product reserves, processing, and manufacturing industries. Consider transferring the capacity of areas with excess agricultural processing capacity to areas with weak agricultural processing capacity by means of asset circulation, a contractual reserve of production capacity, etc., and optimize the allocation of resource elements.

*6.4. Limitations and Further Areas*

Due to the author's limited knowledge, energy, and access to data resources, this study still needs to be improved and developed in the following aspects: (1) The article only includes the primary agricultural products imported by China, such as grain, raw meat, etc., into the index, but there are many varieties of finished products (such as instant food, biscuits, etc.), and due to the difficulty in obtaining relevant data, they are not included; (2) Although China is not a big exporter of agricultural products, it still exports nearly

USD 17 billion in agricultural products (mainly vegetables) in a year. This part of exported vegetables is calculated into the model, which may have errors compared to the actual situation. However, considering that China has 1.4 billion people, the food calculation error of about USD 12 per capita per year may not be unacceptable.

In the future, further research can be carried out on nutrition-oriented food policy, as well as a comparison of food security policies in different countries under the "Great Food View". Additionally, the causes and solutions of the "Matthew effect" of food security in non-main food production areas can be analyzed and the treatment effect of food security policies on food security levels can be explored.

**Author Contributions:** Conceptualization, Q.J. and Z.R.; Methodology, Q.J. and Z.R.; Software, Z.R.; Formal analysis, Q.J. and Z.R.; Investigation, Z.R. and Z.Y.; Data curation, Q.J. and Z.R.; Writing—original draft, Q.J. and Z.R.; Writing—review & editing, Q.J. and Z.Y.; Supervision, Q.J.; Project administration, Q.J. All authors have read and agreed to the published version of the manuscript.

**Funding:** This research was funded by "National Social Science Foundation of China: 22BGL274", "Modern agricultural system economists of China: CARS-46" and " 2021 Shanghai Philosophy and Social Sciences '14th Five-Year Plan' project: 2021BGL009".

**Institutional Review Board Statement:** Not applicable.

**Data Availability Statement:** Data can be provided upon contacting the authors.

**Conflicts of Interest:** The authors declare no conflict of interest.

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
