# Peer review of "Research on the Construction of China’s Provincial Food Security Evaluation System and Regional Performance—Based on the “Great Food View”"

_agriculture, doi:10.3390/agriculture13061240_

Round 1

Reviewer 1 Report

The subject of the article is interesting but major improvements are needed. Please find below my remarks:

1.       Abstract. The abstract do not respect the requirements. It has to contain a brief description of the research goal, method used and results in accordance with the article title. As far as I have understood, the title is about the construction of an evaluation system.  Is the research aim to propose an evaluation system or to make a current evaluation of the region’s effective status in China?

2.       Introduction. The authors failed to design a background of the research. The research objectives are not stated so that all the article’s content becomes confusing.

3.       Theoretical Analysis and Literature review. It is not very clear which is the delimitation between the theoretical analysis and the objectives of the China’s govenamental policy. They should be presented in different subsections.

4.       The methodology is not enough explained. Which research method was used? What data were collected and from what sources? How they were collected and measured?

5.       The Results section is very confusing in the absence of the research objectives. What the authors have emphasized? The evaluation tool from the theoretical and practical point of view or the results of the authors' evaluation regarding the current status?

6.       In the Conclusion section the contribution of the research should be emphasized. The implications of the research results for theory and practice are not clearly described in the article. Research limits should be mentioned and Future research directions should be proposed.

 In my opinion, the authors have to make major revision to this article.

Author Response

Dear Reviewer:

Thank you very much for your comments! 

All authors of the article.

Reviewer 2 Report

This study has taken a substantial attempt to investigate the Construction of China's Provincial Food Security Evaluation System and Regional Performance ⎯⎯Based on the “Great food view. This research work is productive, and the attempts of the authors are praiseworthy. However, the paper is not free from all the defects. Some minor corrections could improve the quality of this paper. The following are my concerns which need to be corrected by the authors.

Introduction

The authors must mention more clearly the research gap. But in my opinion, it has not been articulated in an appropriate manner. I would urge the authors to refer to the research gap in a more comprehensive manner. Besides, in the introduction section, the authors did not mention the contributions of this study in an explicit way. The authors need to elaborate on this issue.

Literature review (Contextualization)

In this section, the authors have duly mentioned the recent citations. But the authors ought to have mentioned more recent citations to make this section more interesting and comprehensive. Also, in this section the authors must explain the literature gap which motivated the authors to take up such interesting research work.

I would welcome just a few more contemporary works, such as:

Sustainable Entrepreneurship and Marketing Strategy: Exploring the Consumer “Attitude–Behavioural-Intention” Gap in the Sport Sponsorship Context”, chapter in book: Ratten V. (eds) Entrepreneurial Innovation. Studies on Entrepreneurship, Structural Change and Industrial Dynamics. Springer, Singapore

Empowerment and performance in SMEs: Examining the effect of employees' ethical values and emotional intelligence”, chapter in book: A Guide to Planning and Managing Open Innovative Ecosystems, Emerald Publishing Limited, ISBN 9781789734102

METHODOLOGY:

The methodology is of great interest and practical value is to apply this methodology to fields with emerging and well-established fields of academics

Limitation and future scope

In this study the authors are found to not have dealt with any rival model which could have given a scope to the authors to compare the proposed theoretical model with the rival model to ascertain the veracity of the proposed theoretical model. At this stage this lacuna must be considered as a limitation of this study. For this, this is to be mentioned in the limitation section suggesting that future researchers should explore this issue.

Lat but not the least the authors must get the manuscript professionally edited to eliminate the typos and other errors. However, the rest of the paper is found satisfactory. I hope that the authors will focus their attention on rectifying the lacunas mentioned. Good luck to the authors.

the authors must get the manuscript professionally edited to eliminate the typos and other errors. 

Author Response

(The authors gave the same response as above.)

Reviewer 3 Report

An interesting study is presented which aims to investigate the level of equality of different provinces in the Peoples Republic of China in terms of food security. Food production was one of the initial criteria of observation. Dynamic findings have been obtained, reflecting changes in food security over the years.

Some observations and suggestions for the paper.

1. Is it possible to show the share of PRC food imports and its role in meeting food security concerns in core and non-core production areas? 

2. Are PRC food exports a constraint in meeting the food security concerns of the country, and to what extent? 

3. State the limitations of the study

Author Response

(The authors gave the same response as above.)

Reviewer 4 Report

This is an interesting paper focusing on food security concept as represented in China's "Great food view".

There is a need for authors to present a more robust background information focusing on food security globally and that of China in recent time. This paragraphs should set a stage for the China 's food security concept in ", Great food view"

Is there any other countries that have related food security concept like Great food view? It will be fine to start with such programmes or intervention before focusing on that of China.

Kindly give concise explanation of table 2.

You mentioned the use of china's secondary data for this study but no concise methods, procedures and tables to show the results from this data sources.

The authors may need to present concise explanation of the results using relevant and recent references to support or refute the findings in this study. 

Before the conclusion section, kindly present a sub-section with the heading "limitations of the study" and "areas for further research"

Thank you.

Minor editing

Author Response

(The authors gave the same response as above.)

Round 2

Reviewer 1 Report

The authors have made the suggested improvements.

Author Response

(The authors gave the same response as above.)

Reviewer 4 Report

Thank you for revising this manuscript. However, you may not need to merge discussion with limitations of the study. Let discussion section stand alone. You are expected to present a discussion using recent and relevant references to either support or refute the findings in this study. This is conspicuously missing here.

In the revised version of this manuscript, kindly note that you're expected to use coloured parts where corrections were made or use track changes. This will make it easy to spot the corrections made. You can only the response here. 

Thank you.

Okay

Author Response

(The authors gave the same response as above.)
